# Spatial genetic structure of 2009 H1N1 pandemic influenza established as a result of interaction with human populations in mainland China

**Seungwon Kim** [1¤]*, **Margaret Carrel** [1,2], **Andrew Kitchen** [3]

**1** Department of Geographical and Sustainability Sciences, University of Iowa, Iowa City, Iowa, United States of America, **2** Department of Epidemiology, University of Iowa, Iowa City, Iowa, United States of America, **3** Department of Anthropology, University of Iowa, Iowa City, Iowa, United States of America

¤ Current address: Department of Pathology, Johns Hopkins University, Baltimore, Maryland, United States of America
* skim519@jhmi.edu

**Data Availability Statement:** All sequences data are available from the NCBI Influenza Virus Database and GISAID EpiFlu database (Accession numbers are within Supporting Information files).

## Abstract

Identifying the spatial patterns of genetic structure of influenza A viruses is a key factor for understanding their spread and evolutionary dynamics. In this study, we used phylogenetic and Bayesian clustering analyses of genetic sequences of the A/H1N1pdm09 virus with district-level locations in mainland China to investigate the spatial genetic structure of the A/H1N1pdm09 virus across human population landscapes. Positive correlation between geographic and genetic distances indicates high degrees of genetic similarity among viruses within small geographic regions but broad-scale genetic differentiation, implying that local viral circulation was a more important driver in the formation of the spatial genetic structure of the A/H1N1pdm09 virus than even, countrywide viral mixing and gene flow. Geographic heterogeneity in the distribution of genetic subpopulations of A/H1N1pdm09 virus in mainland China indicates both local to local transmission as well as broad-range viral migration. This combination of both local and global structure suggests that both small-scale and large-scale population circulation in China is responsible for viral genetic structure. Our study provides implications for understanding the evolution and spread of A/H1N1pdm09 virus across the population landscape of mainland China, which can inform disease control strategies for future pandemics.

## Introduction

The spread of human influenza A viruses among well-connected and highly populated cities is recognized as the primary process driving viral dissemination over a wide range of geographic regions [1–6]. Such viral diffusion also determines the countrywide geographic patterns of genetic structure of the virus; frequent viral exchange between major urban areas may give rise to high degrees of genetic similarity of viral populations across these regions, resulting in

**Funding:** The authors received no specific funding for this work.

**Competing interests:** The authors have declared that no competing interests exist.

broad-scale geographic patterns of genetic homogeneity of viral populations. Increasing connectivity between populations as a result of rapid development of human transportation networks may also facilitate viral exchange not only from urban to urban areas, but also from urban to local (or less populated regions) and local to local areas [7–10], potentially leading to a countrywide viral mixing. This may be particularly the case for influenza pandemic viruses, such as influenza A/H1N1pdm09 virus that spread globally within a few weeks after its first isolation in Mexico and California in April 2009 [11–14]. The A/H1N1pdm09 virus was highly contagious, particularly during the first year of the pandemic, implying that there might be extensive viral exchange and co-circulation of multiple lineages of the virus over a wide range of geographic regions, which could generate unique spatial patterns of genetic structure over the course of the 2009 pandemic. As yet, however, the spatial genetic structure of the A/H1N1pdm09 virus populations has not been fully investigated in all regions.

The geographic patterns of genetic structure of influenza A viruses, such as the geographic distribution of viral populations and the spatial extent to which each lineage of influenza A virus forms genetic clusters across space, provide insight into the underlying mechanisms of evolution and spread of the virus across the human population landscapes [15–17]. Hemagglutinin (HA) and neuraminidase (NA) are two major glycoproteins on the surface of influenza A viruses, which are associated with entry and release in the virus life cycle. These two glycoproteins are the primary targets of the human immune response to the viruses, implying that both genes evolve as a result of the human-virus interaction [18–20]. This suggests that identifying the spatial genetic structure of HA and NA genes helps to improve our understanding of how influenza A viruses interact with humans to adapt to, evolve, and spread across human population landscapes.

The scarcity of specific geographic locations associated with sequenced influenza samples in public databases is one of the major challenges for such studies. Since its emergence in North America, however, a considerable number of influenza A/H1N1pdm09 viral genetic sequences with specific sampling locations have been collected in mainland China during the pandemic period. The availability of this new data provides a unique opportunity to investigate the spatial patterns of virus genetic structure at finer geographic scales, which allows for disentangling the dynamic interaction between humans and the influenza viruses across the landscape [16, 21–23].

In this study, we investigated geographic heterogeneity of genetic structure of HA and NA genes of the A/H1N1pdm09 virus isolated in mainland China during the first year of the 2009 pandemic. Despite the fact that the virus emerged over a decade ago in 2009, the spatial patterns of genetic differentiation of the virus, as well as the aspect of spatial scale in mainland China, are still poorly understood. To address this gap, we examined the spatial genetic structure of the A/H1N1pdm09 viruses at multiple geographic scales using phylogenetic and Bayesian clustering analyses. We analyzed the genetic sequence data of the virus, along with their sampling locations, to identify the geographic patterns of genetic clusters of the viruses in mainland China. This study aimed to investigate the following questions: 1) how genetically divergent were the A/H1N1pdm09 viruses circulating in mainland China during the first year of the pandemic, 2) what were the spatial extent and scale of genetic populations of the virus across the human population landscape of mainland China, and 3) how were the viral populations distributed as a result of dynamic interaction between humans and the viruses? Determining the spatial genetic structure of A/H1N1pdm09 viruses in the early phase of the pandemic can illuminate whether this form of pandemic influenza followed typical patterns of viral diffusion, with genetic homogeneity as a result of high population connectivity, or with highly localized patterns of spatial genetic structure driven by the strong circulation within the communities. In addition, the spatial genetic structure of the virus allows for identifying the

linkage between the human population dynamics and the spread of viral population distribution. Better understanding of spatial patterns of the spread of influenza A/H1N1pdm09 virus will help us develop effective intervention strategies for future influenza pandemics, as well as other airborne infectious viruses, such as SARS-CoV-1.

## Methods

### Sequence samples

Nucleotide sequence datasets for HA and NA genes of the A/H1N1pdm09 virus isolated between April 2009 and August 2010 in mainland China were obtained from the NCBI Influenza Virus Database (IVDB) (HA: 275, NA: 273 –accession date: Oct 29, 2020) and GISAID EpiFlu database (HA: 646, NA: 646 –accession date: Oct 29, 2020) [24, 25]. The datasets were combined and duplicate isolates were removed. Additionally, only sequences that included district-level geographic locations were retained for the study. After removing entries with incomplete sequences in the coding region, the remaining HA and NA gene sequences were aligned in Muscle (v 3.8.31) [26]. Maximum likelihood (ML) phylogenetic trees for HA and NA genes were reconstructed using the general time reversible (GTR) + Gamma substitution model in IQ-TREE (v 2.2.0) [27]. The resulting ML trees were used to identify temporal outliers in TempEst (1.10.4) [28], and those outliers were subsequently removed. Sampling locations were geocoded to the centroid of the district-level boundaries using Google Maps API. The final datasets include 413 sequences for the HA gene and 406 sequences for the NA gene, along with the district-level locations and assignment into one of seven geographic regions (Central, East, North, Northeast, Northwest, South, and Southwest regions) in mainland China (Fig 1), as well as sequence sampling dates (HA: Aug 16, 2009 –Aug 4, 2010, NA: Sep 1, 2009 –Aug 8, 2010) obtained from the sequence metadata (S1 Table).

### Phylogenetic analysis

Temporal phylogenetic trees of HA and NA genes in A/H1N1pdm09 virus were reconstructed to estimate viral evolutionary rates, lineage diversification through time, and time to the most recent common ancestor (tMRCA) of viral lineages (S1 and S2 Figs). Nucleotide sequences of HA and NA genes, and their sampling dates were used to infer phylogenetic trees using a Bayesian Markov Chain Monte Carlo (MCMC) approach in the BEAST software package (v1.10.4) [29]. The HKY+G substitution model selected from the maximum likelihood test in MEGAX [30], the uncorrelated lognormal relaxed molecular clock model [31], and exponential population demographic model were used for the reconstruction of phylogenetic trees [29, 32]. Three independent runs of 100 million generations were performed and sub-sampled every 10,000 generations to ensure MCMC convergence. Tree and log files were combined with a burn in of 10 million samples per run using LogCombiner (v1.10.4). Convergence of parameters was analyzed using Tracer (v1.7.1) [33] and effective sample size (ESS) for all posteriors were greater than 200, indicating sufficient mixing of MCMC chains and no significant autocorrelation in the posterior sample. The tree logs for HA and NA were summarized as maximum clade credibility (MCC) trees in TreeAnnotator (v1.10.4) and visualized in FigTree (v1.4.3) [34].

ML phylogenetic trees with 1,000 bootstrap replications were reconstructed in IQ-TREE (S3 and S4 Figs) to calculate dissimilarity matrices of patristic distances between the genetic sequences of HA and NA genes, the sum of branch length between pairs of samples in the phylogenetic tree [27, 35]. The genetic sequences of A/California/04/2009 (accession number—HA: FJ966082, NA: FJ966084) were added as outgroup for the purpose of rooting, and then

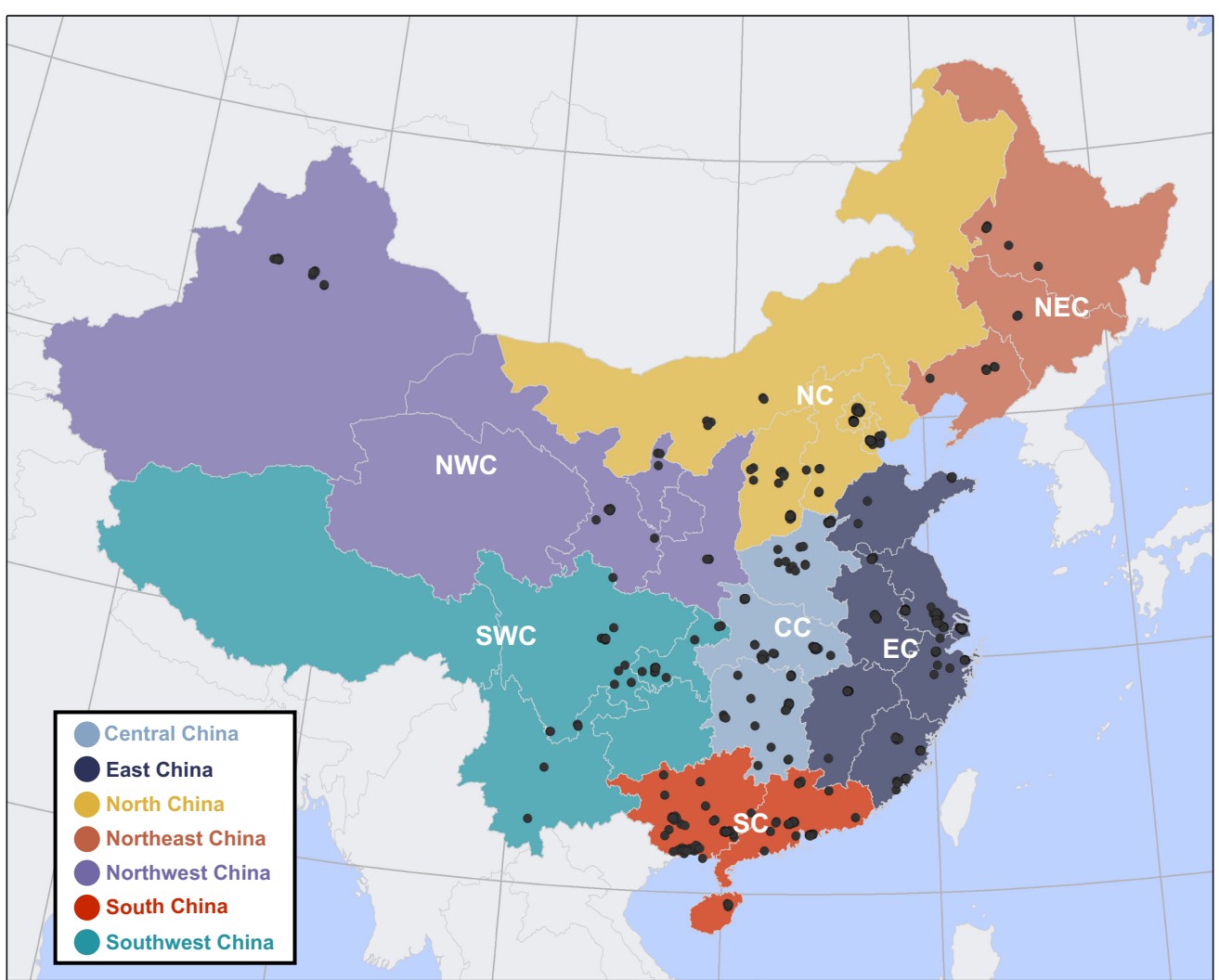

**Fig 1. Sampling locations of genetic sequences of the A/H1N1pdm09 virus (black dots) and seven geographic regions of mainland China.**

removed from the trees. Dissimilarity matrices were generated using *ape* package (v 4.0) in R [36].

## Spatial extent of genetic populations

Partial Mantel correlograms were used to analyze the spatial extent and scale of genetic structure of the A/H1N1pdm09 virus and for identifying the role of human population landscape in the spatial genetic structure of the virus in mainland China. A Mantel correlogram is a non-parametric statistical tool to illustrate the association between two distance matrices, such as genetic distance and geographic distance [37, 38]. The Mantel correlogram classifies pairwise dissimilarity of genetic distances into several geographic distance lags and then calculates a Mantel statistic within each lag. A partial Mantel correlogram, an extension of the Mantel cor-relogram, can utilize three distance matrices simultaneously: a genetic distance matrix as a response variable and two matrices for explanatory variables (e.g., geographic and temporal distances). The partial Mantel correlogram can be used to capture the association between

genetic and geographic distances while accounting for potential confounders, such as temporal trends. Statistical significances within each lag were calculated using a permutation test (n = 10,000). Partial Mantel correlograms were constructed using *ecodist* package [39] in R.

Two different geographic distance matrices were generated to identify the spatial scale of genetic structure of the HA and NA genes: Euclidean distance and indexed distance matrices. Euclidean distance, the classical geographic distance measure in ecological studies, was used to investigate the correlation between genetic and geographic distances. Human population distributions can be considered as landscapes to human influenza viruses: the geographic distribution of human populations, their movements, and population characteristics may determine the spatial patterns of genetic structure of the A/H1N1pdm09 virus. In particular, the population landscape of China is highly heterogeneous; large and densely populated cities are located along the east coast, while the population size and density dramatically decline moving westward. The complexity of human population landscapes in mainland China suggests that the spatial genetic structure of viral diversity might not conform to isolation by distance (IBD) patterns, whereby viruses closer in geographic space are more similar than those further apart in geographic space, which is commonly observed in wild animals, fungi, or plants along a gradient of environmental landscapes [40–42]. The partial Mantel correlograms with Euclidean distance were used to identify the spatial extent of genetic differentiation of the HA and NA genes across the highly heterogeneous human population landscape in mainland China. Euclidean distances between district-level sampling locations were measured using ArcMap 10.7 [43].

Meanwhile, the degrees of genetic similarity between pairs of the virus may vary across different administrative membership; high levels of genetic similarity between viruses are expected within small geographic regions (i.e., community, neighborhood, or district), while such patterns may become less obvious at larger geographic scales (i.e., prefectural and province). The indexed distance measure was designed to capture the genetic spatial structure of the virus at each level of administrative divisions in China. In addition, this measure controls for the impact of different area size across the same levels of administrative divisions that might be unduly influential in the traditional Euclidean measures of space. For instance, two locations may be far apart in geographic space but still belong to the same administrative unit and have greater interactions because of this shared membership than with other sites that are geographically closer but administratively further. The indexed distance measure will thus allow us to identify the degree of genetic similarities across the different spatial scales across mainland China while overcoming the potential confounding impacts of Euclidean distance. The indexed distance matrices for the HA and NA genes were generated based on the sampling addresses in the following ways: 1) if two viral isolates were sampled in the same district-level region, 2) if two viral isolates were sampled in the same prefectural-level region but not in the same district-level region, 3) if two viral isolates were sampled in the same province-level regions but not in the same prefectural-level region, 4) if two viral isolates were sampled in the same geographic region, 5) if two viral isolates were sampled in the different geographic regions.

## Principal component analysis

The geographic differentiation of genetic structure of A/H1N1pdm09 virus by seven geographic regions of mainland China (Fig 1) were investigated using a principal component analysis (PCA) approach. A PCA approach allows us to reduce the dimensions of single nucleotide polymorphism (SNP) frequency data by seeking principal components (PCs); the first PC is the summary of frequency of SNPs that maximize the variance of the projected data, and

the rest of PCs are orthogonal to the first PCs accounting for the residual variance in the data [44, 45]. Two matrices of SNPs frequencies for each gene were generated and used for the PCA. The first two PCs with the highest eigenvalues were used for visualizing the genetic characteristics of viruses by seven geographic regions of mainland China on to two-dimensional space. We also generated 95% inertia ellipses to visualize the genetic differentiation of the A/H1N1pdm09 virus by seven geographic regions. The PCA was performed using *ade4* package in R [46].

## Bayesian clustering analysis

While Mantel correlograms summarize the spatial patterns of genetic structure of the HA and NA genes, this method cannot illustrate the distribution and genetic differentiation of viral populations across space. Bayesian clustering analysis uses the genetic sequence data to calculate the probability that each individual sequence belongs to pre-defined genetic population groups, which allows for classification of the genetic samples into subpopulations and visualization of the probability of the population memberships for each individual sequence in space. By mapping the population structure of the viral samples onto geographic space, the spatially heterogeneous genetic structure of the HA and NA genes can be illustrated across mainland China.

For the clustering analysis, we first identified the optimal number of genetic clusters (K) using an iterative approach in the STRUCTURE software package [47]. Log-likelihood statistics for each value of K, ranging from 1 (all sequences consist of a unique genetic cluster) to 30 (30 genetically heterogeneous groups), were calculated to identify the optimal number of clusters. For the model setting in STRUCTURE, non-admixture model, assuming that each individual is originated from one of K populations, was used to calculate the proportion of clustering membership assigned to each cluster for individual sequences. Further, we allowed allele frequencies in our sequences to be correlated. Five independent runs of 500 thousand MCMC steps with a burn in of 50 thousand steps for each iteration were obtained, combined, and plotted. Two different approaches, non-parametric Wilcoxon test and the ad hoc quantity ($\Delta K$), were used to determine the optimal number of genetic clusters using the combined logs. It is acknowledged that log-likelihoods of K would plateau or slightly continue to increase, and a high variance between runs would be observed as K reaches the true value—the optimal number of subpopulations [48, 49]. Based on these two criteria, the optimal values of K for the HA and NA genes were visually determined from the log-likelihood plots. The $\Delta K$ plots, the second order rate of change of the log-likelihood across Ks, were further used to confirm the optimal Ks identified in the log likelihood plots of the HA and NA genes [50]. After we identified the optimal Ks for each gene, we conducted ten independent runs of two million MCMC steps with a burn in of one million steps at the optimal Ks identified in the previous steps to obtain the membership probabilities for each individual sequence (membership coefficient–q value). Ten independent logs of the membership coefficients were averaged using CLUMPP software [51] and the membership coefficient matrices (q-matrix) of the HA and NA genes were obtained. The probabilities of genetic memberships of each gene were visualized using an Inverse Distance Weighted (IDW) interpolation tool with a power of 2 and 12 minimum number of neighbors to highlight local genetic structure of the virus while capturing global patterns of the genetic structure across mainland China. Contour lines ranging from 0.1 to 0.6 by 0.1 interval were generated and the q-values greater than 0.6 were highlighted to identify the boundaries of high q-values for each genetic cluster. Pairwise Fst values among provinces, measuring the degrees of genetic differentiation of viral populations, were calculated to support our finding of geographic distribution of genetic clusters of the A/H1N1pdm09 virus

using *hierfstat* package in R [52]. The membership probability surfaces for the HA and NA genes were generated in ArcMap 10.7 using the world administrative boundaries shapefile obtained from the World Bank Data Catalog (https://datacatalog.worldbank.org/).

## Results

### Phylogenetic analyses

Phylogenetic trees of the HA and NA genes of A/H1N1pdm09 virus were reconstructed to understand the evolutionary dynamics of the viruses sampled in mainland China during the first year of the pandemic (Aug 2009—Aug 2010). Branches of phylogenies were color coded by year of isolation, revealing that the viruses were not divergent across the year but intermingled across time in the same clades (Fig 2, S1 and S2 Figs). This is a typical pattern observed in phylogenies of pandemic viruses, demonstrative of an explosive increase in genetic diversity of both genes during the early stages of the pandemic. The mean substitution rates of the HA and NA genes were $5.607 \times 10^{-3}$ [95% HPD: $4.66 \times 10^{-3}$–$6.64 \times 10^{-3}$] and $4.97 \times 10^{-3}$ [95% HPD: $3.84 \times 10^{-3}$–$6.14 \times 10^{-3}$] substitutions per site per year, respectively (Table 1). These rates were higher than those from the early phase of the pandemic (HA: $3.67 \times 10^{-3}$, NA: $3.65 \times 10^{-3}$) [53], but not significantly different from results over a longer study period [54].

### Geographic scales of spatial genetic structure

Partial Mantel correlograms based upon two different measures of geographic distance (Euclidean distance and indexed distance) for the HA and NA genes are presented in Fig 3. Statistically significant Mantel r values within each distance lag are symbolized as filled circles,

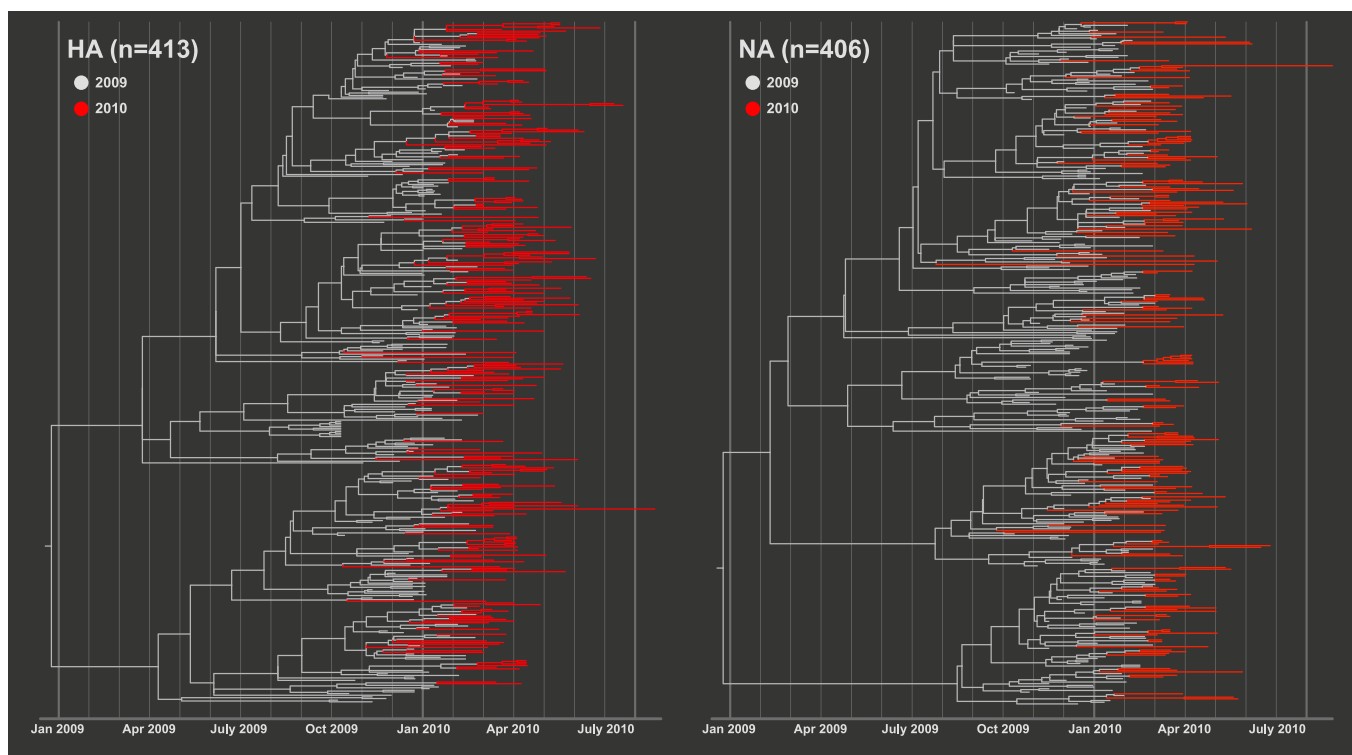

**Fig 2. Bayesian temporal phylogenetic trees of the HA and NA genes in the A/H1N1pdm09 virus sampled in mainland China during the pandemic.** Branches were colored in grey (2009) or red (2010) based on year of viral isolation.

**Table 1. Mean substitution rates and mean tMRCA estimates for the HA and NA gene segments of the A/H1N1pdm09 virus in mainland China.**

| Gene | Mean substitution rate (95% HPD) | Mean tMRCA of root height (95% HPD) |
|------|----------------------------------|-------------------------------------|
| HA | 0.005607 | 2008.970 (Dec 21, 2008) |
|    | [0.4659–0.6638] | [2008.700 (Sep 13, 2008) - 2009.210 (Mar 18, 2009)] |
| NA | 0.4974 | 2008.965 (Dec 19, 2008) |
|    | [0.3842–0.6138] | [2008.659 (Aug 29, 2008) - 2009.248 (Apr 1, 2009)] |

while values that lack significance are represented as hollow circles. Surprisingly, partial Mantel correlograms of the HA and NA genes from Euclidean distance matrices generally present the IBD patterns, where the genetic distance is positively correlated with geographic distance, although such patterns are more evident in the HA than the NA gene. Of note, the Mantel r value around the 2,000km distance lag in the NA gene indicates a significant positive correlation, but this pattern was not seen in the HA gene.

Meanwhile, partial Mantel correlograms with the indexed distance measure illustrate the correlation between genetic sequences across different administrative memberships. The partial Mantel correlogram plot with indexed distance measure (Fig 3) exhibits high positive correlations among samples of the HA gene in the first, second, and third classes, indicative of frequent and strong gene exchanges within the same district, prefecture, and province level regions in mainland China during the period. The Mantel r statistic of HA genes in the fifth

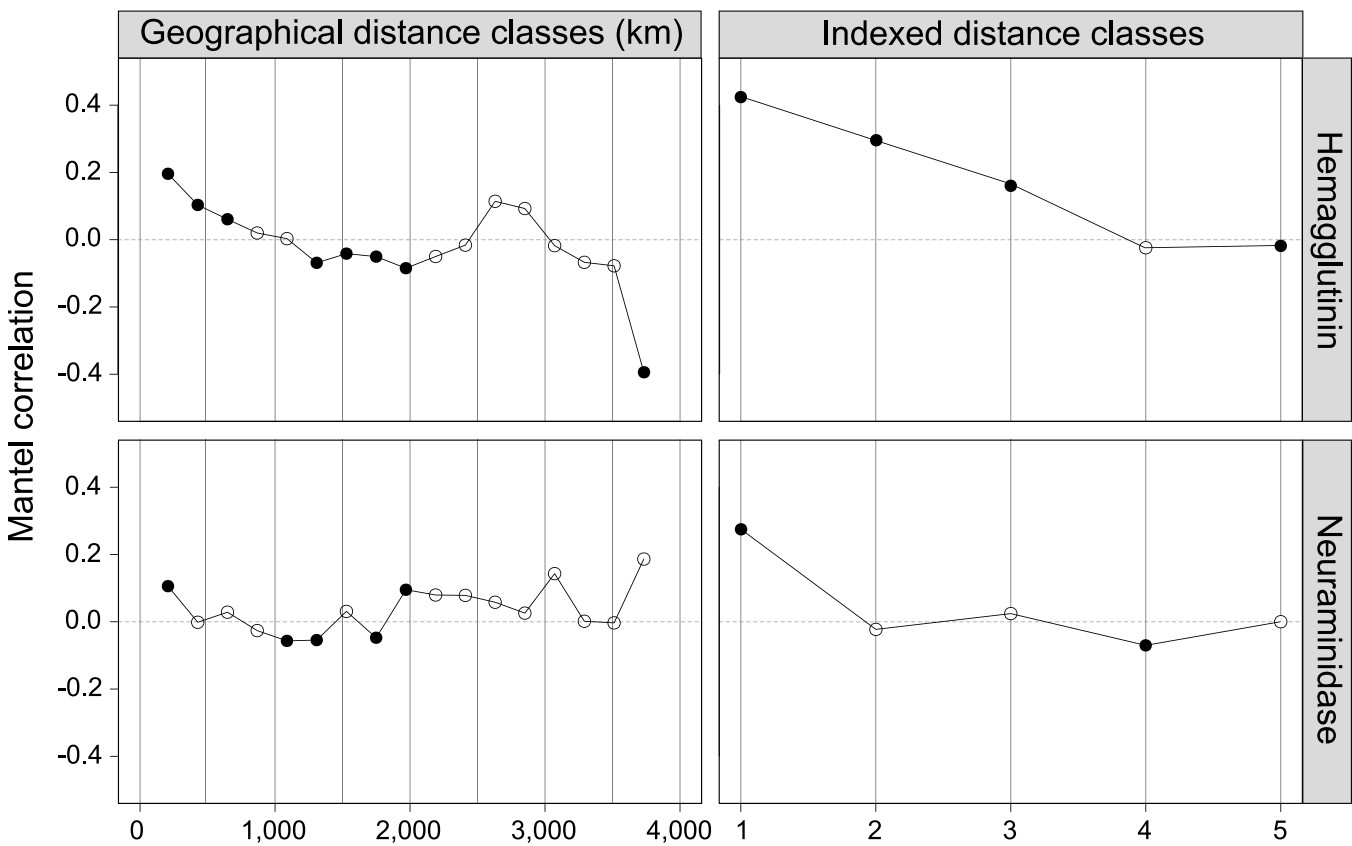

**Fig 3. Mantel correlograms of the HA and NA genes in the A/H1N1pdm09 virus sampled in mainland China during the pandemic.**

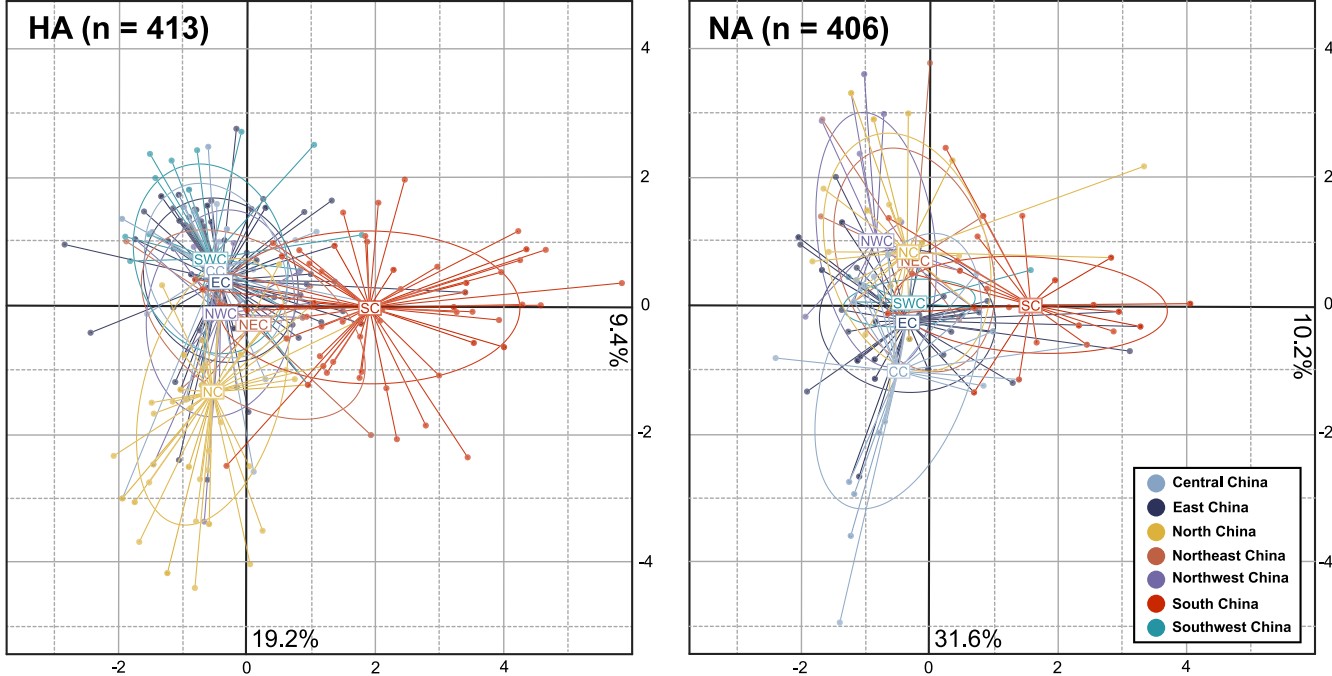

**Fig 4. The scatterplots of the first two PCs from the PCA of the HA and NA genes of the A/H1N1pdm09 virus, color-coded according to seven geographic regions of mainland China.**

class was negative and statistically significant, indicating the presence of genetic differentiation in the HA gene across the seven geographic regions in mainland China. The Mantel r value in the first and fourth class were significant, while the second, third, and fifth classes in NA genes were statistically not significant, implying the absence of genetic structure at these spatial scales. The scatterplots of PCA of the HA and NA genes support these findings, indicating that the geographic patterns of genetic differentiation across the seven geographic regions were more pronounced in HA than NA genes (Fig 4). Collectively, high levels of genetic similarity between HA and NA genes within the same district, prefecture, and provincial level regions imply the presence of the genetic clusters at these spatial scales, while the genetic differentiation of the HA genes across seven geographic regions was observed.

## Bayesian clustering analyses

The geographic patterns of the distribution of subpopulations of HA and NA genes of the A/H1N1pdm09 virus were investigated using an individual-level Bayesian clustering approach. First, we identified the optimal numbers of genetic clusters for each HA and NA gene using the non-parametric Wilcoxon test and Δ K plots. The log-likelihoods plot of the HA (S5 Fig) gradually increased from K = 1, exhibited a large variance at K = 3 followed by reaching its plateau of the log-likelihood statistic from K = 4. The ad hoc quantity (Δ K) indicates clear peak at K = 4, supporting our finding of the optimal K for HA genes. The log-likelihood values of the NA gene reached its plateau at K = 2 with a large variance of log-likelihood statistics between K = 1 and K = 2. In the ad hoc quantity plot of the NA gene, the Δ K was maximum at K = 2, supporting our finding of the optimal number of subpopulations of the NA genes from the log-likelihood plot. Taken together, the optimal number of subpopulations of the HA genes was larger than these of the NA genes, suggesting that the HA genes were more genetically diverged than the NA genes during the first year of the pandemic.

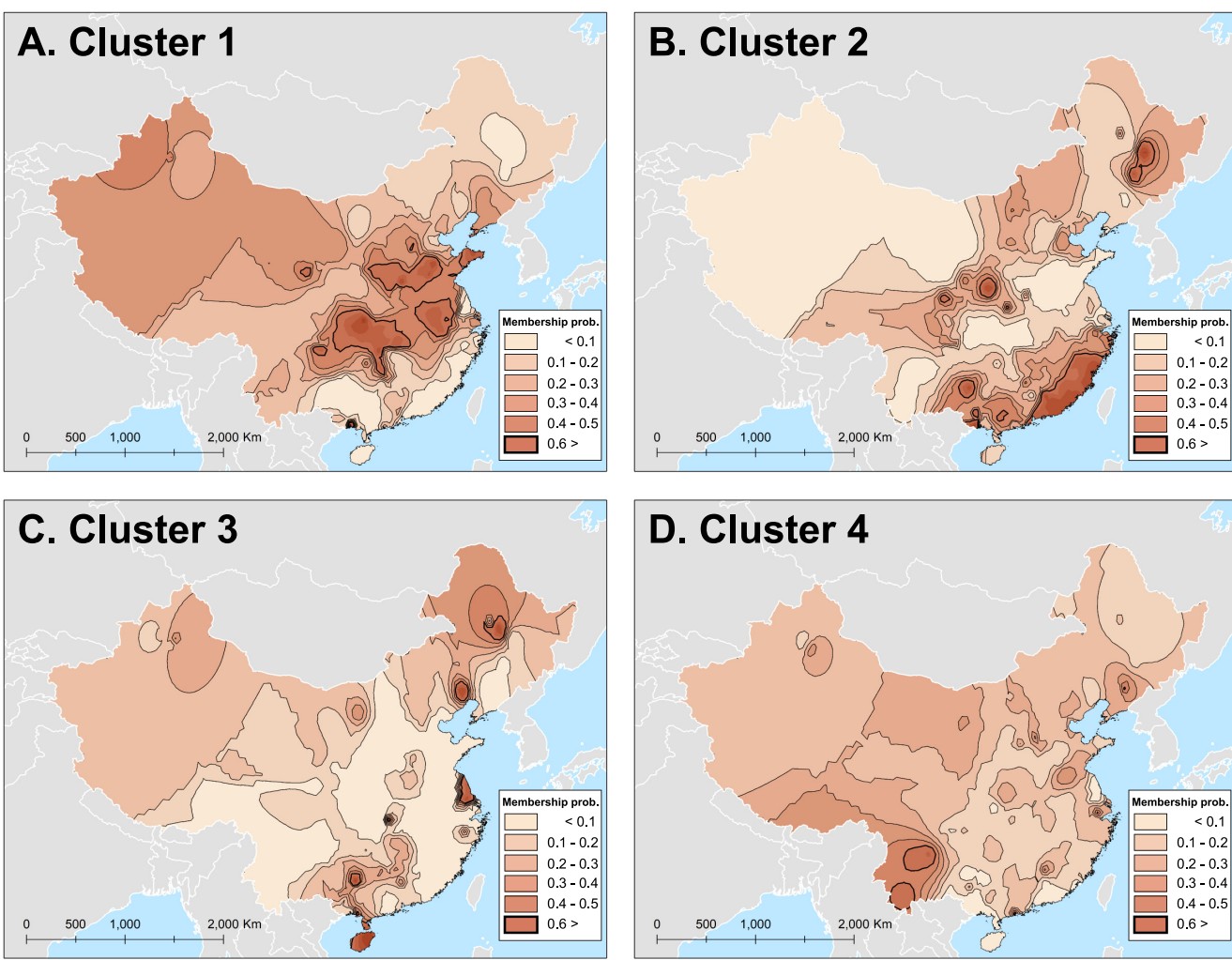

**Fig 5. The membership probability surfaces of HA genes of A/H1N1pdm09 virus generated using an Inverse Distance Weighted interpolation.** Thick contour lines represent the boundaries of high q-values greater than 0.6.

The probability of genetic memberships assigned to each individual sequence of the HA and NA genes of the A/H1N1pdm09 virus (S2 and S3 Tables and S6 Fig) were visualized using the IDW interpolation to investigate the spatial patterns of the distribution of genetic subpopulations of the HA and NA genes. Fig 5A illustrates that the HA genes with high membership probabilities of Cluster 1 were widely distributed across East and Central China. Provinces which contain contour lines of the IDW with q values greater than 0.6 include Sichuan, Chongqing, Hunan, Hubei, Anhui, Henan, Shanxi, and Shandong. Notably, both global and localized spatial genetic clusters are apparent in the map of Cluster 1. Specifically, there is a broad cluster spanning from the Shandong peninsula in the east coast to the inner provinces (Fig 5A) and high degrees of population genetic similarity among these provinces were identified from pairwise Fst values (S4 Table and S7 Fig). Meanwhile, locally confined subpopulations (genetic demes) were observed in several regions, mainly in Gansu, Sichuan, and Guangxi. In contrast, the spatial distribution of HA genes with higher membership probability for Cluster 2 illustrates a more geographically confined pattern that is dominant in South

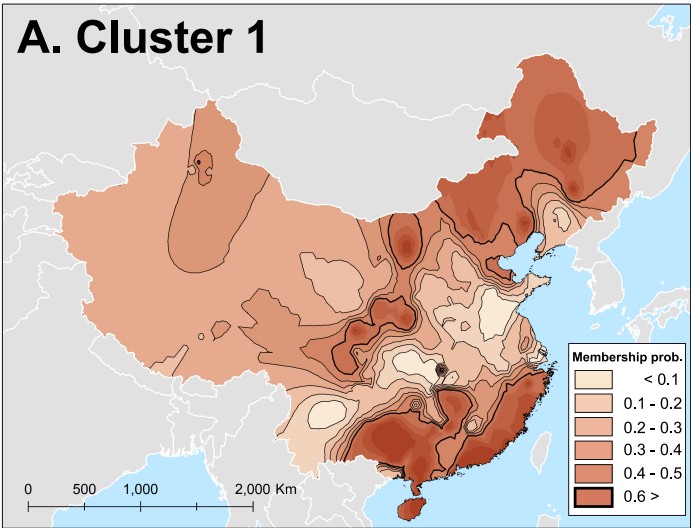
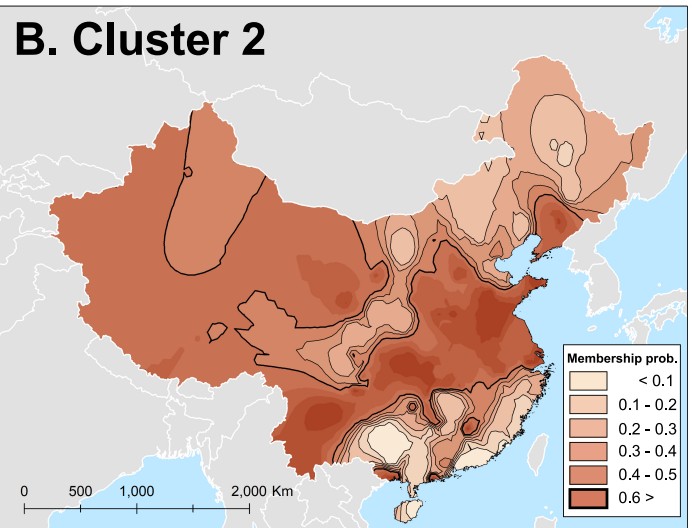

**Fig 6. The membership probability surfaces of NA genes of A/H1N1pdm09 virus generated using an Inverse Distance Weighted interpolation.** Thick contour lines represent the boundaries of high q-values greater than 0.6.

China, particularly in Guangdong and Fujian provinces, although small local subpopulations of with Cluster 2 membership were also found in Heilongjiang, Jilin, Shaanxi, and Guangxi (Fig 5B, S4 Table and S7 Fig). In contrast, genes with majority membership to Clusters 3 and 4 are found in sub-regional clusters. The geographic distribution of HA genes with q values of Cluster 3 greater than 0.6 illustrate locally isolated patterns, observed mainly in Heilongjiang, Zhejiang, Guangdong, Guangxi, and Hainan (Fig 5C). Interestingly, analysis of pairwise Fst between Hainan and other provinces indicated the smallest genetic differentiation between HA genes in Hainan and Heilongjiang (S4 Table and S7 Fig). Lastly, mapping Cluster 4 membership identified geographic clusters with q > 0.6 only in Yunnan province, and no other significant genetic subpopulation was observed in other regions (Fig 5D).

The clustering analysis of NA gene sequences identified only two genetic clusters, as visualized in Fig 6. The genetic sequence samples of NA that were classified as Cluster 1 are mainly distributed across North, Northeast, and South China (Fig 6A), while the NA genes with higher membership probability of Cluster 2 were widely observed in East and Central China (Fig 6B). Although the geographic differentiation of the genetic cluster of the NA genes is evident, the spatial patterns of the genetic clusters were determined by only two subpopulations.

## Discussion

Our study used influenza A/H1N1pdm09 virus HA and NA gene sequences with known sampling locations to investigate the geographic patterns of genetic structure and genetic differentiation of the virus in mainland China during the initial active phase of the 2009 H1N1 pandemic. Due to the highly mobile nature of humans and, by extension, the influenza virus, we expected that broad-scale geographic patterns of genetic population structure might be dominant over localized patterns of genetic clustering, as multiple introductions from major cities into local communities can blur the genetic population structure of the virus by increasing the genetic diversity in local areas (see for example [10]). In addition, only few research investigated the geographic distribution of the genetic mutations or spatial genetic structure of the virus at a national scale [55], and most previous studies have focused on the influenza trends within a single city or province in China [56–59]. Our study, however, highlights the

presence of spatial genetic structure at both national and local scales across mainland China that emerged during the initial stages of the pandemic.

Partial Mantel correlograms with Euclidean distance indicate the presence of IBD patterns in the genetic structure of HA and NA genes, clearly showing the positive correlation between genetic and geographic distances. Given that influenza can travel long distances via multiple transportation networks, frequent viral migration among major municipalities and provinces, as well as gene flow from those regions to local communities, is expected. This may lead to genetic homogeneity among distant but well-connected populations, while multiple introductions of more than two lineages into the local areas may blur the genetic structure of viral populations at smaller geographic scales [10]. Contrary to our hypotheses, however, the results indicate high degrees of genetic similarity among the HA and NA genes sampled within short geographic distances, while spatial genetic differentiation was found across mainland China. These patterns are clearly shown in the partial Mantel correlograms with indexed distance and the scatterplots of PCA, indicating high degrees of genetic similarity between HA and NA genes within the same district regions, and the HA genes in the same prefectural and provincial regions, and the statistically significant genetic differentiation of HA genes across the seven geographic regions of China (Figs 3 and 4). The small genetic demes in the maps of genetic clustering (e.g., Fig 5A) further support our finding of the strong local circulation of the virus during the study period.

The genetic homogeneity of the HA and NA genes at small geographic scales suggests that founder effects might take place at these spatial scales [60, 61]. Though there were likely multiple introductions of different lineages into local areas, only a few successful lineages of the A/H1N1pdm09 virus would become dominant, while the majority of introduced lineages would fail to persist in local or regional populations. Although extensive viral migration, presumably from East China, is supported by geographical clustering (Fig 5A, S7 Fig and S4 Table), country-wide viral exchange and gene flow do not appear to be a predominant driver in establishing a nation-wide pattern of spatial population structure. There appears to be regional differences in which lineage became dominated during the first year of the H1N1 pandemic in China. However, we cannot completely rule out potential bias in these spatial patterns, as only a small proportion of influenza cases were genotyped, and these sequences may not be sufficient to represent the genetic diversity within the regions. Therefore, further study is needed to investigate the genetic characteristics of A/H1N1pdm09 virus at small geographic scales.

Interestingly, the IBD patterns of Mantel correlogram of NA gene are more ambiguous than those of HA genes, particularly in terms of the significant positive Mantel r values near the 2,000km distance lag. These unclear IBD patterns in the Mantel correlogram of NA gene may be because the NA gene was under weaker selection pressure and not as genetically differentiated as the HA gene. Although both HA and NA genes determine the antigenic immune profile of influenza A viruses, the greater neutralizing potential of anti-HA antibodies imposes stronger selection pressures on the HA gene [62–65]. In addition to being a target for host immune neutralization, the HA gene synthesizes the spike-like receptor binding glycoprotein on the surface of the virus which allows the virus to bind to the host cells, and thus plays an important role in determining the infectivity of viral particles. Although the NA protein is also highly mutable and under diversifying positive selection in response to anti-NA antibody pressure [62, 66], positive selection pressure is generally stronger on the HA gene due to the highest concentration of epitopes in the HA1 sub-domain of the HA protein [54, 67], which is the least conserved segment of influenza virus and the major target of human immunity against the virus. This is further supported by the estimates of the mean substitution rates of the HA and NA genes during the study period (Table 1). The mean substitution rates of NA were estimated as generally lower than those documented for the HA gene, implying that the NA genes

were genetically less differentiated. Taken together, the NA genes in the A/H1N1pdm09 virus were less genetically diverged over the first year of the pandemic, thereby the genetic differentiation of NA genes by geographic distance might be less obvious, particularly within the range of 0–2,500km and as shown in the scatterplots of PCA (Fig 4).

We found broad-scale spatial patterns of distribution of the HA genes with high probability of Cluster 1 membership (Fig 5A). These patterns were also evident in pairwise Fst estimates, indicating low genetic differentiation among provinces in East and Central China (S4 Table and S7 Fig). These patterns may be attributable to the movement of migrant workers from East China to rural areas in Central and Western China. In particular, the Yangtse River Delta, one of the major industrial regions in China, covers a wide range of geographic regions including the provinces of Jiangsu, Anhui, Zhejiang, and Shanghai, which combined account for more than 20% of rural migrant workers in China [68–70]. Many of the migrant workers that travel to the east originate in inner provinces with large rural populations, such as Sichuan, Henan, Anhui, and Shandong [69]. This large population of migrant workers is also involved in the periodic population movements from east to west during the Lunar New Year holiday when the flu is most prevalent and incidence is high. These geographic patterns may suggest that the viruses might maintain circulation among human populations in East China and then spread to local communities in Central and Southwest China, giving rise to the formation of local genetic clusters of the virus in these regions. It should be noted, however, that the periodic movement of migrant workers may not be a sole driver of the spatial patterns of Cluster 1 subpopulation distributions, but other factors may play a role in the formation of spatial genetic structure in mainland China, such as climate, socioeconomic, and demographic characteristics in each region, or viral introductions from other regions outside mainland China via international air travel [71–73]. Previous studies analyzed the role of human movements during the Lunar New Year holiday in the spread of human infectious diseases across China (e.g., influenza, STDs, and SARS-CoV-1) [74–78]. However, this study only provides descriptive interpretations of the results, thus the association between human migrations and genetic structure was not statistically tested. Therefore, further model-testing and data collection are necessary to identify the specific demographic and environmental forces driving these spatial patterns of genetic structure across East and Central China.

The HA genes with high probability of Cluster 2 membership were observed mainly in South China during the study period (Fig 5B). Cultural landscapes in South China are characterized by highly populated urban areas, frequent domestic and international trade, extensive human travel between neighboring countries, and a large number of seasonal migrant workers from other provinces [79–81]. In particular, over the past three decades, Guangdong province remains the largest destination of rural migrant workers, accounting for 44% of the total population in the province in 2004 [69]. The population inflow from rural areas has dramatically increased the population size and density of cities in Guangdong province, such as Shenzhen, one of the destination cities for rural migrant workers in China [82]. These rural migrant workers often live in poor and overcrowded housing conditions, with low incomes, generally low awareness of disease prevention, and poor immunization status, which makes these populations more vulnerable to respiratory infectious diseases and widespread viral circulation [82–85]. These characteristics of the cultural landscape of South China might reinforce the local circulation of the virus among the large number of human populations of South China [86, 87].

It is still questionable, however, that the HA genes with high probability of Cluster 2 membership were geographically confined to South China and failed to spread beyond the region. South China has historically been proposed to serve as a human influenza A epicenter responsible for at least two influenza pandemics in the previous century, A/H2N2 in 1957 and A/H3N2 in 1968 [88, 89], implying a high potential of viral strains circulating in South China to

have more antigenicity than those circulating in other regions. The results were opposite to what we expected, however, indicating that the strains in South China were relatively geographically confined in these regions, presumably because of less competitive antigenicity or infectivity than the viral strains that were found in East and Central China. It is still unclear the relationship between genetic/antigenic characteristics of the virus in South China and their geographical constraints acting on their spatial distribution. Therefore, identifying the prevalence of influenza A viruses among migrant workers in South China and local persistence of the virus over time is required to understand the role of human population landscape in the geographic patterns of genetic structure in the region.

The map of Cluster 3 membership and pairwise Fst identified high degrees of genetic similarity of the HA genes among Heilongjiang in North China, Zhejiang in East China, and Guangdong, Guangxi, and Hainan in South China (Fig 5C and S7 Fig). Interestingly, analysis of pairwise Fst between Hainan and other provinces indicated that the genetic differentiation between Hainan and Heilongjiang was the smallest, implying frequent viral exchange between two distant regions. Notably, Northeast China has a northerly continental monsoon climate with long and cold winters, lasting from November to March with an average daily high temperature below 27˚F. Because of this seasonal variation, it is common for people in Northeast China to vacation in tropical places to escape the harsh winters of this region. Hainan province, in particular, is one of the popular destinations for Northeast Chinese tourists during the Lunar New Year holiday [90], which coincides with the period when flu is most prevalent. Despite the distance between these two geographic regions, a large volume of returning tourists from South to Northeast China after the Lunar New Year vacations might be sufficient to establish long-distance viral transmission, resulting in a high degree of genetic similarity between viruses from these two regions. Meanwhile, local genetic clusters in Guangxi, Guangdong, and Hunan may form due to the geographic proximity to Hainan. However, a more formal phylogeographic approach is required to identify the gene flow of the A/H1N1pdm09 virus across these regions [54, 91, 92].

The spatial genetic structure of the NA sequences in the cluster maps was evident, presenting high levels of genetic similarity between NA gene sequences sampled in North and South China, and viral population that extends from East to Southwest China. However, we observed only two genetic subpopulations of in our sample of NA gene sequences, due to low degree of genetic differentiation. This would seem unusual, as anti-NA antibodies in humans help reduce both the replication of the virus and its virulence [93], though it is possible selection for immune escape within the HA gene overwhelmed similar selection pressures in the NA gene during the early stages of the pandemic.

This study was conducted using genome sequences of the A/H1N1pdm09 virus isolated only in mainland China. This geographically constrained sample may fail to capture viral exchanges between China and other countries, and thus under-estimate the degree to which independent introductions of virus might affect our inferences of geographic structure of viral gene sequence diversity. Furthermore, the short sampling period of only one year (August 2009 –August 2010) may not be sufficient to examine the evolution and geographic patterns of genetic differentiation of the virus in space. Moreover, early viral sequences sampled before August 2009 that cover the first wave of the pandemic in early spring were excluded from this study due to their lack of associated specific geographic locations. Earlier samples might prove important, as although geographic patterns of genetic differentiation might not exist at the earliest phase of the pandemic, these viral samples may provide insight into how A/H1N1pdm09 virus was first introduced to and how it subsequently spread through mainland China. Lastly, spatially and temporally uneven sampling should be addressed to avoid potential bias in the outcome derived from over/under sampled provinces in future studies.

The geographic patterns of the genetic structure of the pandemic influenza viruses presented in this study imply the strong association between human population connections and the spread of the virus. The IBD patterns of spatial genetic structure of the A/H1N1pdm09 viruses were clear, suggesting efficient viral circulation at smaller geographic scales (i.e., districts, prefectural, and provincial regions) and genetic differentiation at large scales. The results also highlight the presence of both global and local patterns of spatial genetic structure of A/H1N1pdm09 virus HA and NA genes. Periodic population movements from provinces along the east coast to inner provinces may contribute to broad-scale geographic patterns of genetic structure, while localized genetic subpopulations may imply that viral transmission from highly populated urban areas to local communities as well as local to local areas was also significant during the first year of the pandemic.

Our findings are expected to provide the basis for surveillance and intervention strategies before/after the emergence of new pandemic viruses. Active influenza surveillance during the Lunar New Year holiday may help control the viral transmission driven by periodic population movements, while monitoring the viruses circulating in densely populated areas along the east coast of China may allow for the identification of the antigenic variants in a timely manner. Identifying the geographic distribution of place-specific predominant influenza viral strains will be helpful for developing influenza vaccines and efficient vaccination plans that maximize the efficiency for disease control. Lastly, further investigation of landscape factors, such as climate, human transportation networks, and socioeconomic characteristics, is necessary to improve our knowledge about the underlying drivers of the genetic differentiation of human influenza viruses in preparation of public health strategies for future pandemics.

## Supporting information

**S1 Fig. Bayesian temporal phylogeny of HA genes of A/H1N1pdm09 virus with sequence names.**
(PDF)

**S2 Fig. Bayesian temporal phylogeny of NA genes of A/H1N1pdm09 virus with sequence names.**
(PDF)

**S3 Fig. Maximum likelihood phylogeny of HA genes of A/H1N1pdm09 virus.** Internal nodes with a bootstrap support of 50% or greater (1,000 replications) are indicated.
(PDF)

**S4 Fig. Maximum likelihood phylogeny of NA genes of A/H1N1pdm09 virus.** Internal nodes with a bootstrap support of 50% or greater (1,000 replications) are indicated.
(PDF)

**S5 Fig. The summary of log-likelihood statistics and the estimates of ΔK over K = 1–30 from STRUCTURE.**
(PDF)

**S6 Fig. The bar plots of the membership probabilities of HA and NA genes of A/H1N1pdm09 virus by province by geographic region in mainland China.**
(PDF)

**S7 Fig. The maps of pairwise Fst estimates of HA genes between Shandong, Guangdong, and Hainan and their top 5 closest provinces in mainland China.**
(PDF)

**S1 Table. A list of isolate names, sources, isolate ID, and collection date used in the study.**
(CSV)

**S2 Table. Summary of the membership coefficients (K = 4) of HA genes calculated in CLUMPP software.**
(CSV)

**S3 Table. Summary of the membership coefficients (K = 2) of NA genes calculated in CLUMPP software.**
(CSV)

**S4 Table. Pairwise Fst estimates of HA genes of A/H1N1pdm09 virus among provinces in mainland China.**
(XLSX)

**S5 Table. Pairwise Fst estimates of NA genes of A/H1N1pdm09 virus among provinces in mainland China.**
(XLSX)

## Author Contributions

**Conceptualization:** Seungwon Kim, Margaret Carrel, Andrew Kitchen.

**Data curation:** Seungwon Kim.

**Formal analysis:** Seungwon Kim.

**Methodology:** Seungwon Kim, Andrew Kitchen.

**Supervision:** Margaret Carrel.

**Validation:** Andrew Kitchen.

**Visualization:** Seungwon Kim.

**Writing – original draft:** Seungwon Kim.

**Writing – review & editing:** Margaret Carrel, Andrew Kitchen.

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
