## [Decision Letter · Decision Letter 0]

6 Nov 2022

PONE-D-22-15803Spatial genetic structure of 2009 H1N1 pandemic influenza established as a result of interaction with human populations in mainland ChinaPLOS ONE

Dear Dr. Kim,

Thank you for submitting your manuscript to PLOS ONE. After careful consideration after a too long review process with a search with more than 15 referees,  we feel that it has merit but does not fully meet PLOS ONE’s publication criteria as it currently stands. Therefore, we invite you to submit a revised version of the manuscript that addresses the points raised during the review process.

ACADEMIC EDITOR:The referee and I have found some technical issues that need to be addressed:A recombination analysis that could affect the results of your analysisThere is missing information in the Materials and Methods sectionSome Figures legends are missing as well as details in the phylogenetic treesReview the Discussion Section because some statements are too bold and need some clarificationPlease ensure that your decision is justified on PLOS ONE’s publication criteria and not, for example, on novelty or perceived impact.

We look forward to receiving your revised manuscript.

Kind regards,

Cecilio López-Galíndez

Academic Editor

PLOS ONE

Journal Requirements:

2. We note that Figures 1, 5 & 6 in your submission contain [map/satellite] images which may be copyrighted. All PLOS content is published under the Creative Commons Attribution License (CC BY 4.0), which means that the manuscript, images, and Supporting Information files will be freely available online, and any third party is permitted to access, download, copy, distribute, and use these materials in any way, even commercially, with proper attribution. For these reasons, we cannot publish previously copyrighted maps or satellite images created using proprietary data, such as Google software (Google Maps, Street View, and Earth). For more information, see our copyright guidelines: http://journals.plos.org/plosone/s/licenses-and-copyright.

 a. You may seek permission from the original copyright holder of Figures 1, 5 & 6 to publish the content specifically under the CC BY 4.0 license. 

Reviewers' comments:

Reviewer's Responses to Questions

**Comments to the Author**

1. Is the manuscript technically sound, and do the data support the conclusions?

Reviewer #1: Partly

2. Has the statistical analysis been performed appropriately and rigorously? 

Reviewer #1: No

3. Have the authors made all data underlying the findings in their manuscript fully available?

Reviewer #1: Yes

4. Is the manuscript presented in an intelligible fashion and written in standard English?

Reviewer #1: Yes

5. Review Comments to the Author

Reviewer #1: The study by Kim, Carrel and Kitchen investigates the spatial genetic structure of 2009 H1N1 influenza in China. I found the topic of the study interesting but there are several aspects that should be considered. First, I missed an analysis of recombination in these data. Recombination could affect genetic diversity, genetic structure and also phylogenetic tree reconstructions. Second, there is a lot of missing information in methods and results. For example, the bar plot about the populations estimated for every location with the software STRUCTURE is missing, also how the phylogenetic trees were rooted and a bootstrap analysis to evaluate the statistical support of the internal nodes of the reconstructed trees (if one aims to show a geographic structure, clades from different regions should appear clearly separated by high bootstrap values). Legends of figures of the main text are also missing and also stats about the clustering (Fig 5). I found the discussion pretty risky, with too much speculation. I was wondering if the geographic origin of the expansion could be identified and how was the spatiotemporal expansion. Also if the virus evolved following (or not following) a molecular clock. The specific comments follow below. I recommend major revisions.

Main comments

I missed an analysis of genetic recombination in these data. Recombination can affect genetic diversity, genetic structure and also the phylogenetic tree reconstructions and inferences derived. See for example,

https://www.ncbi.nlm.nih.gov/pmc/articles/PMC1461297/

https://pubmed.ncbi.nlm.nih.gov/20124027/

I was wondering if the geographic origin of the virus expansion in China could be predicted, for example analyzing genetic diversity or ancestral sequence reconstruction, among other methods. And also, if possible, could be interesting to understand the spatiotemporal dynamics of the expansion from the genetic data, including geographic direction and speed of the expansion. Not only phylogenetic reconstructions can be useful for this, there are other software that considers geographic constraints, i.e.,

https://pubmed.ncbi.nlm.nih.gov/29044712/

https://www.tandfonline.com/doi/full/10.1038/s41426-018-0185-z

The study should include additional previous studies about the spatial genetic structure of 2009 H1N1 influenza in some regions of China and compare with their results when possible, for example,

https://journals.plos.org/plosone/article?id=10.1371/journal.pone.0028027

https://www.ncbi.nlm.nih.gov/pmc/articles/PMC3086218/

https://www.pnas.org/doi/10.1073/pnas.1921186117

I noted that crucial information is missing in the study.

The study mentions the analysis of the population structure with the software STRUCTURE, however there is not any bar plot showing the estimated populations for every location. For example see Fig 5 in https://www.nature.com/articles/s41467-018-05257-7

How the phylogenetic trees were rooted? This is fundamental to understand the evolutionary trajectories over time. It should be clarified in detail.

I missed a bootstrap analysis to evaluate the statistical support of the internal nodes of the reconstructed phylogenetic trees. If one aims to show a geographic structure, clades from different regions should be separated by nodes with high bootstrap values.

Also, it could be useful for readers to visualize the tree with clades for every geographic region (or for every cluster of Fig 5) presented with different colors.

I missed a principal component analysis (PCA) for detecting clusters, note that PCA is frequently used for this purpose.

Also, presenting FST estimates between populations from the same and from different clusters could be useful.

I found the discussion quite risky, with too much speculation trying to connect possible human migration events with the observed genetic structure. Many factors can affect the genetic structure, including local interactions, virus transmission and evolution, etc. I recommend being more cautious.

Minor comments

I was wondering if the virus evolved following (or not following) a molecular clock. This can be explored with BEAST.

I could not find the legends explaining what is shown in the figures of the manuscript.

6. PLOS authors have the option to publish the peer review history of their article (what does this mean?). If published, this will include your full peer review and any attached files.

Reviewer #1: **Yes: **Miguel Arenas

---

## [Author Response · Author response to Decision Letter 0]

23 Jan 2023

Dear Dr. Arenas and Editors, 

Thank you for providing us the opportunity to revise the manuscript. We appreciate the time and effort that Dr. Arenas and editors have dedicated to providing valuable feedback. Please find the revised manuscript and our answers based on the reviewer’s comments below. 

Editor: Copyright of Figure 1, 5 and 6

We generated our figures using the world administrative boundaries shapefile (vector format for spatial data) obtained from the World Bank. The source of the shapefile is noted in the revised manuscript from line 246 to 247, and their license information and policy can be found in the link below:

https://datacatalog.worldbank.org/public-licenses?fragment=cc

Main comment 1

Comment: I missed an analysis of genetic recombination in these data. Recombination can affect genetic diversity, genetic structure and also the phylogenetic tree reconstructions and inferences derived.

Answer: Recombination is one of the main drivers of the evolution of influenza A viruses, particularly with regard to the process of reassortment, which is the process of recombination between genomic segments. Homologous recombination within segments is relatively (very) rare in the evolution of influenza A viruses (Nelson & Holmes 2007; Boni et al. 2008) and it is very unlikely that intra-segment recombination might take place during our one-year of study period (Aug 2009 – Aug 2010). Furthermore, recombinants in our sequence datasets would be identifiable as outliers in our root-to-tip analyses, as they would exist at the tips of long branches in the tree, and thus be more genetically distant from the root than expected given their sampling time. Therefore, we did not perform a recombination analysis for our alignments which is standard for the field (e.g., Nelson et al. 2008; Rambaut et al. 2008; Bedford et al. 2015), as there is no indication that one was necessary.

Main comment 2

Comment: I was wondering if the geographic origin of the virus expansion in China could be predicted, for example analyzing genetic diversity or ancestral sequence reconstruction, among other methods. And also, if possible, could be interesting to understand the spatiotemporal dynamics of the expansion from the genetic data, including geographic direction and speed of the expansion. 

Answer: We agree with Dr. Arenas that identifying the geographic origins (as there are most certainly more than one entry point into such a large and diverse country) would be quite interesting. Indeed, previous studies have suggested that the origins of the A/H1N1pdm09 virus in China were the major cities of mainland China, such as Beijing and Shanghai (Bin et al., 2009; Liu et al., 2011), which is expected given typical pandemic virus epidemiology. Unfortunately, this type of combined phylogeographic analysis is beyond the scope of this study, as it would require many additional sequences from outside of China to contextualize the sequences from within China. This is because there are likely multiple introductions to China, and each of these introductions would require identification by including sequences from intervening geographic regions. Given this, a large phylogeographic study could be performed in a subsequent paper after we have resolved the movement of pandemic H1N1 within China during the early stages of the epidemic.

Main comment 3

Comment: The study should include additional previous studies about the spatial genetic structure of 2009 H1N1 influenza in some regions of China and compare with their results when possible.

Answer: While previous A/H1N1pdm09 studies in mainland China are often aspatial, analyzing the influenza trends within a single city or province, this study investigated the global patterns of the genetic structure across mainland China. Integrating findings from the local influenza trends and the genetic structure at a national scale would be challenging, as the spatial patterns of genetic characteristics of the virus may vary across the different spatial scales. Previous studies of 2009 H1N1 influenza were included in the first paragraphs of discussion. 

Main comment 4

Comment: The study mentions the analysis of the population structure with the software STRUCTURE, however there is not any bar plot showing the estimated populations for every location.

Answer: The revised manuscript contains bar plots of HA and NA genes from the analysis of population structure in STRUCTURE, the estimated populations by province and by seven geographic regions (Supplementary figure 6). 

Main comment 5

Comment: How the phylogenetic trees were rooted? This is fundamental to understand the evolutionary trajectories over time. It should be clarified in detail.

Answer: We used ‘A/California/04/2009’ as an outgroup sequence for rooting the ML phylogenetic trees, which is now clearly indicated in the methods. The Bayesian trees estimated in BEAST are automatically rooted using a molecular clock model informed by sampling times that imposes directionality on the tree.

Main comment 6-1

Comment: I missed a bootstrap analysis to evaluate the statistical support of the internal nodes of the reconstructed phylogenetic trees. If one aims to show a geographic structure, clades from different regions should be separated by nodes with high bootstrap values.

Answer: We included ML phylogenetic trees of HA and NA genes with the bootstrap supports of inner nodes greater than 50% as supplementary information. Importantly, we do not over-analyze the structure of the tree, but rather the pattern of pairwise distances between sequences along the tree, which is more robust to interpretation than the tree alone.

As we illustrated in maps, PCA plots, and Mantel’s correlograms, the geographic differentiation of genetic structure at regional or province scales is not apparent, and the findings from ML phylogenetic trees are consistent with these results. Only few internal nodes close to the tips have the bootstrap support greater than 50%, and the geographic differentiation of sequences by geographic region is not clear in phylogenetic trees as well (i.e., there are few monophyletic clades from specific regions of China). It is only in the analysis of the total matrix of pairwise distances that geographic patterns become apparent.

Main comment 6-2

Comment: It could be useful for readers to visualize the tree with clades for every geographic region (or for every cluster of Fig 5) presented with different colors.

Answer: The membership probabilities of HA obtained in STRUCTURE are not categorical but continuous, ranging from 0 to 1, thus it may not be appropriate to annotate these membership values in phylogenetic trees. Instead, we included ML trees with tips colored by geographic regions in supplementary information (Sup Fig 3 and 4) to see if the sequences from the same provinces tend to be in the same clades with the higher bootstrap supports. 

Main comment 7-1

Comment: I missed a principal component analysis (PCA) for detecting clusters, note that PCA is frequently used for this purpose.

Answer: The PCA plots were included as supplementary information and the PCA analysis is now fully described in the methods section. 

Main comment 7-2

Comment: Also, presenting FST estimates between populations from the same and from different clusters could be useful.

Answer: We calculated pairwise Fst estimates between provinces and reported as tables in supplementary information. In addition, maps of Fst estimates 1) between Hainan and top 5 closest provinces, 2) between Guangdong and top 5 closest provinces, and 3) between Shanghai and top 5 closest provinces were included as supplementary information. 

Main comment 8

Comment: I found the discussion quite risky, with too much speculation trying to connect possible human migration events with the observed genetic structure. Many factors can affect the genetic structure, including local interactions, virus transmission and evolution, etc. I recommend being more cautious.

Answer: We revised the discussion sections to be more cautious about our interpretation of our findings. The Fst maps included as supplementary information may support our speculation about gene flows between populations. 

Minor comment 1

Comment: I was wondering if the virus evolved following (or not following) a molecular clock. This can be explored with BEAST.

Answer: We used TempEst (v 1.5.3) to see if the evolutionary rates of HA and NA followed expected clock-like evolution, and it did.

Minor comment 2

Comment: I could not find the legends explaining what is shown in the figures of the manuscript.

Answer: The titles of legends were updated. 

References

Bedford T, Riley S, Barr IG, Broor S, Chadha M, Cox NJ, Daniels RS, Gunasekaran CP, Hurt AC, Kelso A, Klimov A, Lewis NS, Li X, McCauley JW, Odagiri T, Potdar V, Rambaut A, Shu Y, Skepner E, Smith DJ, Suchard MA, Tashiro M, Wang D, Xu X, Lemey P, Russell CA. 2015. Global circulation patterns of seasonal influenza viruses vary with antigentic drift. Nature 523:217-220.

Bin C, Xingwang L, Yuelong S, Nan J, Shijun C, Xiayuan X, Chen W, National Influenza A Pandemic (H1N1) 2009 Clinical Investigation Group. Clinical and epidemiologic characteristics of 3 early cases of influenza A pandemic (H1N1) 2009 virus infection, People’s Republic of China, 2009. Emerging infectious diseases. 2009 Sep;15(9):1418.

Boni MF, Zhou Y, Taubenberger JK, Holmes EC. 2008. Homologous recombination is very rare or absent in human influenza A virus. Journal of Virology 82:4807-4811.

Liu Y, Wang W, Li X, Wang H, Luo Y, Wu L, Guo X. Geographic distribution and risk factors of the initial adult hospitalized cases of 2009 pandemic influenza A (H1N1) virus infection in mainland China. PloS one. 2011 Oct 12;6(10):e25934.

Nelson MI, Holmes EC. 2007. The evolution of epidemic influenza. Nature Reviews Genetics 8:196-205.

Nelson MI, Edelman L, Spiro DJ, Boyne AR, Bera J, Halpin R, Ghedin E, Miller MA, Simonsen L, Viboud C, Holmes EC. 2008. Molecular epidemiology of A/H3N2 and A/H1N1 influenza virus during a single epidemic season in the United States. PLoS Pathogens 4:e1000133.

Rambaut A, Pybus OG, Nelson MI, Viboud C, Taubenberger JK, Holmes EC. 2008. The genomic and epidemiological dynamics of human influenza A virus. Nature 453:615-619.

---

## [Decision Letter · Decision Letter 1]

21 Feb 2023

PONE-D-22-15803R1Spatial genetic structure of 2009 H1N1 pandemic influenza established as a result of interaction with human populations in mainland ChinaPLOS ONE

Dear Dr. Kim,

Thank you for submitting your manuscript to PLOS ONE. After careful consideration, we feel that it has merit but does not fully meet PLOS ONE’s publication criteria as it currently stands. Therefore, we invite you to submit a revised version of the manuscript that addresses the points raised during the review process.

We look forward to receiving your revised manuscript.

Kind regards,

Yanpeng Li, Ph.D.

Academic Editor

PLOS ONE

Journal Requirements:

Reviewers' comments:

Reviewer's Responses to Questions

**Comments to the Author**

1. If the authors have adequately addressed your comments raised in a previous round of review and you feel that this manuscript is now acceptable for publication, you may indicate that here to bypass the “Comments to the Author” section, enter your conflict of interest statement in the “Confidential to Editor” section, and submit your "Accept" recommendation.

Reviewer #1: All comments have been addressed

Reviewer #2: (No Response)

2. Is the manuscript technically sound, and do the data support the conclusions?

Reviewer #1: Yes

Reviewer #2: (No Response)

3. Has the statistical analysis been performed appropriately and rigorously? 

Reviewer #1: Yes

Reviewer #2: (No Response)

4. Have the authors made all data underlying the findings in their manuscript fully available?

Reviewer #1: Yes

Reviewer #2: (No Response)

5. Is the manuscript presented in an intelligible fashion and written in standard English?

Reviewer #1: Yes

Reviewer #2: (No Response)

6. Review Comments to the Author

Reviewer #1: This new version of the study is much more clear and detailed. I do not have additional comments and thus I recommend accept it for publication.

Reviewer #2: Kim et al. utilized phylogenetic and Bayesian clustering methods to analyze genetic sequences of the A/H1N1pdm09 virus in mainland China, with a focus on district-level locations. Their objective was to examine the spatial genetic structure of the virus across human population landscapes. The Discussion section is also very detailed. Overall, the article contains a relatively comprehensive level of detail. However, there are a few minor issues that require resolution in order to meet the requirements for publication.

There are several issues that need to be further clarified by the authors:

1. Assuming the authors have some knowledge of the research context, but it may be helpful to provide more background information on the study design, methods, and objectives to help readers understand the research question and the significance of the results.

2. Lines 425-457. The paragraph suggests that the movement of migrant workers is a potential driver of the observed genetic structure in influenza viruses. However, the evidence for this claim is somewhat circumstantial. The authors should clarify the strength of the evidence supporting this hypothesis and acknowledge other potential drivers of genetic structure.

3. Is there any other similar research indicating that other viruses, especially RNA viruses, have transmission characteristics similar to the results of this study?

7. PLOS authors have the option to publish the peer review history of their article (what does this mean?). If published, this will include your full peer review and any attached files.

Reviewer #1: **Yes: **Miguel Arenas

Reviewer #2: No

---

## [Author Response · Author response to Decision Letter 1]

1 Apr 2023

Dear reviewers and Editors, 

We would like to express our gratitude for the comments and feedbacks on our manuscript. We appreciate the time and effort the editors and reviewers have taken to review our work, and we have carefully considered all of the suggestions and recommendations. Please find the revised manuscript and our answers based on the second reviewer’s comments below.

Comment 1. Assuming the authors have some knowledge of the research context, but it may be helpful to provide more background information on the study design, methods, and objectives to help readers understand the research question and the significance of the results.

Answer: We appreciate your suggestion. We revised the last paragraph of introduction to make clear about our study objectives, methods, and implication of the study results. 

Comment 2. Lines 425-457. The paragraph suggests that the movement of migrant workers is a potential driver of the observed genetic structure in influenza viruses. However, the evidence for this claim is somewhat circumstantial. The authors should clarify the strength of the evidence supporting this hypothesis and acknowledge other potential drivers of genetic structure.

We should admit that this is one of the limitations of this study. As we were not able to obtain the population migration data during the first year of the 2009 pandemic in mainland China, the association between the periodic movements of migrant workers and the spatial patterns of genetic structure was not statistically tested. Instead, we included other previous studies investigating these relationships between migration and transmission in China, while clarifying the limitation of the study in the paragraph. As you suggested, we mentioned other potential factors driving viral spreading in the same paragraph. 

Comment 3. Is there any other similar research indicating that other viruses, especially RNA viruses, have transmission characteristics similar to the results of this study?

Previous studies analyzed the geographic patterns of viral transmission and the association between human migrations and the spread of human infectious diseases in China, such as COVID-19 or STDs (e.g., Smith, 2005; Chen, et al., 2020; Wen et al., 2020; Zhan, et al., 2020). However, each virus has their own ecological and evolutionary characteristics to maintain circulation among human populations, thus spatial genetic structure of these virus may vary by viruses, environmental, or population characteristics (e.g., population immunity, population density, connectivity, and migration patterns). Moreover, these patterns may change over time as viruses evolve and adapt to enhanced human immunity from previous infections or vaccination, as we have seen in the COVID-19 cases. Therefore, we found that comparing the spatial patterns of the genetic structure of different viruses was not informative, instead, we focused on influenza A/H1N1pdm09 viruses in mainland China in this study. 

Reference

Chen, Huijie, et al. "Correlation between the migration scale index and the number of new confirmed coronavirus disease 2019 cases in China." Epidemiology & Infection 148 (2020).

Charu, Vivek, et al. "Human mobility and the spatial transmission of influenza in the United States." PLoS computational biology 13.2 (2017): e1005382.

Smith, Christopher J. "Social geography of sexually transmitted diseases in China: Exploring the role of migration and urbanisation." Asia Pacific Viewpoint 46.1 (2005): 65-80.

Wen, Y., Wei, L., Li, Y., Tang, X., Feng, S., Leung, K., ... & Mei, S. (2020). Epidemiological and clinical characteristics of coronavirus disease 2019 in Shenzhen, the largest migrant city of China. MedRxiv, 2020-03.

Zhan, C., Tse, C. K., Fu, Y., Lai, Z., & Zhang, H. (2020). Modeling and prediction of the 2019 coronavirus disease spreading in China incorporating human migration data. Plos one, 15(10), e0241171.

---

## [Editor Report · Decision Letter 2]

6 Apr 2023

Spatial genetic structure of 2009 H1N1 pandemic influenza established as a result of interaction with human populations in mainland China

PONE-D-22-15803R2

Dear Dr. Kim,

We’re pleased to inform you that your manuscript has been judged scientifically suitable for publication and will be formally accepted for publication once it meets all outstanding technical requirements.

Kind regards,

Yanpeng Li, Ph.D.

Academic Editor

PLOS ONE

---

## [Editor Report · Acceptance letter]

8 May 2023

PONE-D-22-15803R2 

Spatial genetic structure of 2009 H1N1 pandemic influenza established as a result of interaction with human populations in mainland China 

Dear Dr. Kim:

I'm pleased to inform you that your manuscript has been deemed suitable for publication in PLOS ONE. Congratulations! Your manuscript is now with our production department. 

Kind regards, 

on behalf of

Prof. Yanpeng Li 

Academic Editor

PLOS ONE